# Extending Single Cell Bioprinting from Femtosecond to Picosecond Laser Pulse Durations

**DOI:** 10.3390/mi12101172

**Published:** 2021-09-29

**Authors:** Jun Zhang, Yasemin Geiger, Florian Sotier, Sasa Djordjevic, Denitsa Docheva, Stefanie Sudhop, Hauke Clausen-Schaumann, Heinz P. Huber

**Affiliations:** 1Lasercenter, Department of Applied Sciences and Mechatronics, Munich University of Applied Sciences, Lothstrasse 34, 80335 Munich, Germany; zhang@hm.edu (J.Z.); ygeiger@hm.edu (Y.G.); sdjordje@hm.edu (S.D.); 2Center for Applied Tissue Engineering and Regenerative Medicine CANTER, Munich University of Applied Sciences, Lothstrasse 34, 80335 Munich, Germany; stefanie.sudhop@hm.edu (S.S.); hauke.clausen-schaumann@hm.edu (H.C.-S.); 3Center for NanoScience, University of Munich, 80799 Munich, Germany; 4Experimental Trauma Surgery, Department of Trauma Surgery, University Regensburg Medical Centre, Am Biopark 9, 93053 Regensburg, Germany; denitsa.docheva@klinik.uni-regensburg.de; 5InnoLas Photonics GmbH, Justus-von-Liebig-Ring 8, 82152 Krailling, Germany; florian.sotier@hm.edu

**Keywords:** laser-induced forward transfer (LIFT), film-free LIFT, single-cell bioprinting, tissue engineering, femtosecond laser-based bioprinting, picosecond laser-based bioprinting

## Abstract

Femtosecond laser pulses have been successfully used for film-free single-cell bioprinting, enabling precise and efficient selection and positioning of individual mammalian cells from a complex cell mixture (based on morphology or fluorescence) onto a 2D target substrate or a 3D pre-processed scaffold. In order to evaluate the effects of higher pulse durations on the bioprinting process, we investigated cavitation bubble and jet dynamics in the femto- and picosecond regime. By increasing the laser pulse duration from 600 fs to 14.1 ps, less energy is deposited in the hydrogel for the cavitation bubble expansion, resulting in less kinetic energy for the jet propagation with a slower jet velocity. Under appropriate conditions, single cells can be reliably transferred with a cell survival rate after transfer above 95% through the entire pulse duration range. More cost efficient and compact laser sources with pulse durations in the picosecond range could be used for film-free bioprinting and single-cell transfer.

## 1. Introduction

Tissue engineering is based on the generation of artificial tissues through a combination of cells and suitable, biocompatible materials to repair, replace, or regenerate diseased or injured tissue. However, the major challenge in producing functional tissue substitutes ex vivo lies in the replication of the complex and unique tissue microarchitecture of native tissue, consisting of different cell types and extracellular matrix [1,2,3]. Laser-induced forward transfer (LIFT) technology has been used to print biomaterials and mammalian cells onto substrates with almost no loss of bioactivity [4,5,6,7,8,9]. The fast and precise generation of 2D and 3D structures has a great potential for applications in the field of tissue engineering [10,11,12,13]. In a typical setup, bioink is usually coated onto a few tens of nanometer thick metal energy-absorbing layer (EAL) as a thin-film with a thickness of about 100 µm. However, the material of the EAL can be transferred by a ns-laser together with the printed bioink and can contaminate the target structure [14]. To avoid this contamination, protein-based hydrogels together with an ultraviolet (UV) laser have been used for energy absorption. However, this process frequently causes DNA double-strand breaks [15], rendering these techniques potentially toxic or carcinogenic. As an alternative approach, a film-free laser-based bioprinting technique has been developed: Femtosecond (fs) near infrared (NIR) laser pulses focused into a transparent liquid [16,17,18,19] generate a spatially confined optical breakdown near the beam waist, which absorbs a large portion of the laser pulse energy. This results in a rapidly expanding cavitation bubble with high pressure, which propels the biomaterial towards an acceptor surface [20,21]. Thus, high intensities of fs-pulses in combination with non-linear interactions steer the absorption into the volume of a transparent medium. If the photon intensity in the laser focus is sufficient to generate an optical breakdown, neither an inorganic EAL nor an UV-laser source are required for the transfer process.

In our previous work [22,23], we demonstrated that by focusing a 600 fs laser pulse with a wavelength of 1030 nm into a hydrogel underneath a layer of cells, small droplets of hydrogel containing living cells can be ejected from the reservoir and transferred to an acceptor stage. The transferred cells showed a survival rate up to 95% without DNA double strand breaks and maintained their ability to migrate and proliferate even 66 h after the cell transfer. Furthermore, the single cells can be transferred with a position precision of ±11.8 µm. By integrating this film-free cell transfer approach into an inverted epifluorescence microscope, this new approach allowed the precise and efficient selection and positioning of individual mammalian cells from a complex cell mixture (based on morphology or fluorescence) onto a 2D target substrate or a 3D pre-processed scaffold with single cell spatial precision [23]. It was found that the transfer process requires less than 100 µs per cell, manually searching for a cell takes ≈5 s, setting the laser focus another ≈5 s, and positioning of the acceptor surface takes further ≈10 s. To date the technique can be used for single cell studies. Future developments for upscaling to produce e.g., cell niches are already planned. This approach will thus enable the precise, fast, and cell-friendly fabrication of cell-chips, organs-on-a-chip, 3D organoids, and ultimately of functional tissue substitutes.

In previous work we used a commercial fs laser source for the generation of the optical breakdown, which is currently used for eye surgery in a few ten thousand units in clinics and medical offices, with typical output parameters of a few microjoules (µJ) pulse energy and a few 100 fs pulse duration [24,25,26]. These laser sources are mostly based on the Nobel-prize-winning technique of chirped pulse amplification (CPA) [27], but are usually relative expensive and too large to directly be integrated into an optical microscope. Set-ups with CPA fs-lasers integrated in a microscope commonly result in a footprint of about two m^2^ on an optical table. From literature it is well known that an optical breakdown and a subsequently expanding cavitation bubble can also be generated by pulse durations in the picosecond (ps) range [28]. For ps laser pulse generation CPA is not essential, leading to less expensive, more compact, and easier-to-integrate systems.

The jet dynamics of ps as well as fs pulses was studied by Petit et al. [29] but neither the generation of laminar jets nor bioprinting of cells could be demonstrated. Thus, there is a gap in the existing state of the art for the generation of regular and laminar jet with ps pulses and the demonstration of their bioprinting properties.

In the present study, we therefore investigated the bubble and jet dynamics as well the energetics for laser pulse durations from 600 fs to 14.1 ps. For all studied pulse durations, we could generate laminar jets. With increasing laser pulse duration, we observed less efficient energy conversion and comparable, but slower jet dynamics. In our earlier work the cell survival rate obtained immediately after the transfer showed no significant difference in their ability to migrate and proliferate over a 66 h time scale [22,23]. Thus, to test the principal bioprinting compatibility at different pulse durations, only the cell survival rate 15 min after transfer was determined. Under appropriate conditions, single cells can be reliably transferred with a high survival rate after transfer through the entire pulse duration range. This agrees very well with our previous findings obtained for a laser pulse duration of 0.6 ps [22,23]. Nevertheless, it is mandatory to investigate the long-term cell viability of ps-laser printed cells in a future study.

## 2. Methods

To vary the pulse duration, a commercial femtosecond laser (femto*1030-25-Yb, Innolas Photonics, Krailling, Germany) with a wavelength of 1030 nm and a variable pulse duration from 600 fs to 14.1 ps (Appendix A) was used. The maximum pulse energy of the laser was 25 µJ. For the laser-based transfer setup, the laser pulses were focused through a water immersion objective (HC FLUOTAR L 25×/0.95 W VISIR, Leica, Wetzlar, Germany) into a medium with a density of 1.083 g/mL (histopaque 1083, Sigma-Aldrich, Taufkirchen, Germany) as shown in Figure 1. Due to the lower mass-density, living human mesenchymal stem cells (hMSC, SCP1 cell line) were suspended at the hydrogel surface (Figure 1 left). To visualize and investigate the process dynamics, including cavitation bubble expansion and subsequent jet propagation, pump-probe [30] time-resolved imaging observation was carried out, as described previously [23]. Briefly, the transfer process was illuminated by a 28 ns white-light pulse (Nanolite KL-L, High-Speed Photo-Systeme, Wedel, Germany) and then was captured with an imaging system comprising a microscope objective (Mitutoyo Plan Apo 5×/0.14, Japan), a tube lens (TTL200-A, Thorlabs, Bergkirchen, Germany), and a CCD1 camera (PCO, Pixefly USB, Kelheim, Germany). The electronic delay of the light source was triggered by a photodiode (DET10A/M, Thorlabs, Bergkirchen, Germany) and synchronized by a delay generator (DG645, Stanford Research Systems, CA, USA). For systematic analysis, the time-resolved image at each delay time was repeated at least three times.

To determine the energetics of the ultrafast laser-induced transfer, the mechanical energy of the cavitation bubble was calculated at its maximum size, where the pressure inside the bubble is equal to the pressure of the surrounding liquid [20,31]. Thus, the mechanical energy for the cavitation bubble expansion (i.e., bubble energy) in the hydrogel can be calculated by:(1)$$E_{B}=\frac{4\pi }{3}\left(p_{0}-p_{v}\right)R_{max}^{3}$$
where $p_{0}$ is the hydrostatic pressure and $p_{v}$ the water vapor pressure. For the setup used here, these were 100 kPa and 2.33 kPa at 20 °C, respectively [31,32]. When the expanding bubble is close to the liquid surface, the bubble energy is partly converted into kinetic energy for the jet formation, resulting in a hydrogel jet from the liquid surface towards the acceptor surface.

In a next step the conversion of the incident laser pulse energy into mechanical energy is considered. Therefore, the kinetic energy of the propagating jet is estimated at a constant delay time of 10 µs. It is assumed that the jet shape is a truncated cone with a height h, which is in rest at the hydrogel surface and propagates with the observed jet front velocity v at the tip. The jet volume is approximated as a cylinder with a diameter d, which equals the diameter of the cone at half height h. The average jet velocity is regarded as the half of the jet velocity v at the tip and the hydrogel density ρ is constant (1.083 g/mL). Thus, the approximated kinetic energy $E_{k}$ is given by:(2)$$E_{k}=\frac{\pi }{32}\rho d^{2}hv^{2}$$

To evaluate the survival rate of the transferred cells, the laser process was integrated into an inverted epi-fluorescence microscope (Nikon Ti-E) with a closed incubation chamber (Ibidi, Gräfelfing, Germany), which provides a constant temperature of 37 °C and >90% humidity to incubate the cells prior to and after the transfer. For bioprinting, human mesenchymal stem cells (hMSC, SCP1 cell line) [33] were used in this work.

The preparation of the hydrogel reservoir, the acceptor surface, and the final analysis of the cell-survival rate are described in previous works [22,23] and are briefly summarized here: detached cells were suspended in 400 µL histopaque 1083 (Sigma-Aldrich, Taufkirchen, Germany) and transferred to a reservoir (μ-Dish 35 mm low, Ibidi, Gräfelfing, Germany).

Gelatin (Sigma-Aldrich, Taufkirchen, Germany) was dissolved in phosphate-buffered saline (PBS, Sigma-Aldrich, Taufkirchen, Germany), to a concentration of 10% *w/v* and then coated with a thickness of about 100 µm on a coverslip, which serves as an acceptor for the transferred cells. The gelatin gel becomes liquid under the incubation condition, thus possible mechanical damage induced by the impact of landing cells on the acceptor slide is minimized and considered to be neglectable.

Detection of dead cells in the cell reservoir prior to the laser transfer and on the gelatin coated acceptor surface after printing is described in previous work [23]: propidium iodide ReadyProbes reagent (PI R37108, Thermo Fischer, Germering, Germany) was pipetted into the above-mentioned bioink of hMSCs and gelatin to a final concentration of 2 droplets (~200 µL) per milliliter. Dead cells were then displayed in red under the epifluorescence microscope and only living cells were selected and transferred to the acceptor. After cell transfer, the acceptor surface was incubated for 15 min to allow PI staining. For statistical analysis, at least 17 individual single cells were transferred for each printing parameter.

## 3. Results

To determine the mechanical energy of the cavitation bubble, bubble formation was initiated by focusing a 2 µJ laser pulse about 1 mm below the hydrogel surface (Figure 2a). At this distance from the symmetry-breaking hydrogel-air interface, the cavitation bubble expanded isotropically and collapsed again without a hydrogel jet leaving the reservoir surrounding the cavitation bubble. For a pulse duration of 0.6 ps, the cavitation bubble reached its maximum size at a radius $R_{max}$ of 85 ± 1 µm at a delay time of about 10 µs. Subsequently, the bubble collapsed and expanded again, oscillating a few times, while decreasing in size (Figure 2b). The development of the cavitation bubbles generated by the 1.8, 9.7, and 14.1 ps laser pulses was similar to that generated by the 0.6 ps pulse, as shown in Appendix A, supporting information. Figure 2b–e plots the bubble radius as a function of delay time at various pulse durations. The maximum radius $R_{max}$ of the bubble decreased slightly from 84.2 ± 1.2 µm for 0.6 ps to 81.1 ± 2.3 µm for 1.8 ps to 75.3 ± 2.3 µm for 9.7 ps and to 71.3 ± 2.2 µm for 14.1 ps pulse (Figure 2f), and the corresponding bubble energy decreased accordingly, from 244 ± 10 nJ at 0.6 ps to 149 ± 13 nJ at 14.1 ps (Figure 2g, left *y*-axis).

We assessed the bubble energy conversion efficiency of the laser process by dividing the bubble energy (Equation (1)) with the total incident laser pulse energy of 2 µJ. The so-defined efficiency decreased from 12.2 ± 0.5% for 0.6 ps to 7.4 ± 0.7% for 14.1 ps pulse duration (Figure 2g, right *y*-axis).

To investigate how the jet dynamics during film-free bioprinting depend on the laser pulse duration, a time-resolved study of the jet propagation was carried out as shown in Figure 3a. The laser pulse energy and the focus depth were kept constant at 2 µJ and 52 µm, respectively. This setting allowed a stable ejection of hydrogel without splashing for all pulse durations up to 14.1 ps. The jet fronts versus the delay time after the laser pulse are plotted in Figure 3b, where the slopes of the fits correspond to the jet velocities. In Figure 3b, each data point is the mean of three independent experiments. Error bars correspond to the standard deviations. Over the entire pulse duration range the jet dynamics were comparable to our previous work under identical laser pulse energy and focus depth [23]. The jets were laminar and regular at all pulse durations. However, the jet velocity v decreased from 25.6 ± 0.6 m/s for 0.6 ps to 24.0 ± 0.8 m/s for 1.8 ps, to 17.5 ± 1.0 m/s for 9.7 ps, and 14.3 ± 1.0 m/s for 14.1 ps. The kinetic energy $E_{k}$ was determined at a delay time of 10 µs. The jet kinetic energy decreased from 17 ± 2 nJ at 0.6 ps to 3.0 ± 0.5 nJ at 14.1 ps and the corresponding conversion efficiency dropped from 0.83% to 0.15%, respectively (Table 1). The kinetic energy conversion efficiency displayed a stronger drop with rising pulse duration than the bubble energy conversion efficiency.

To determine the cell survival rate after transfer, the process was performed with the same laser parameters (2 µJ pulse energy and 52 µm focus depth) as for the time-resolved imaging displayed in Figure 3. In this case, all jets developed laminarly straight up to the acceptor surface without splashing. Only living cells were selected and transferred to the gelatin-coated acceptor surface (see Methods section for preparation details). Figure 4 shows bright-field and fluorescence images of transferred cells on acceptor slide. Live cells appear in green, while dead cells appear in red due to PI staining. Table 2 shows the total number of transferred and viable hMSC cells 15 min after the transfer. Almost all printed cells survived after transfer. In the worst case, here at 1.8 ps, 20 out of 21 cells survived after transfer.

For statistical analysis, the cell survival after transfer was determined by varying laser pulse energy from 1.5 µJ to 3 µJ and focus depth from 39 µm to 65 µm at different pulse durations as shown in Appendix A. Over all 16 experiments with different parameters 275 out of 281 cells survived. Such a result would correspond to an average survival rate after transfer of 97.9% and agrees very well with our previous findings obtained for a laser pulse duration of 0.6 ps [22,23].

## 4. Discussion

As shown in Figure 2, by increasing the laser pulse duration from 0.6 to 14.1 ps the maximum cavitation bubble radius and the corresponding mechanical energy of bubble expansion were reduced, indicating that less radiation energy is converted into mechanical energy. Considering the bubble energy at 2 µJ pulse energy, the conversion efficiency to bubble energy dropped from 12.5% at 0.6 ps to 7.5% at 14.1 ps (Figure 2g). In the same pulse duration range the conversion efficiency into kinetic energy of the jet declined even more significantly from 0.83% at 0.6 ps to 0.15% at 14.1 ps (Figure 3c).

The initial pressure of the bubble rapidly decreases, as the bubble expands and the surrounding liquid is displaced and the energy is finally dissipated in the hydrogel [28,31]. When the expanding bubble is generated close to the liquid air interface, the expanding bubble drives a hydrogel jet, which is ejected from the hydrogel reservoir and transferred to the adjacent acceptor surface [34,35]. We suspect pulse-duration-dependent absorption as the main effect for the decline of bubble energy with pulse duration. Furthermore, dissipation effects for the displacement of the surrounding hydrogel should be responsible for the observed further reduced kinetic energy remaining for the jet propagation, resulting in smaller jet velocities at higher pulse durations (Figure 3c).

The absorption in an optical breakdown depends on the pulse duration and the ratio of pulse energy and breakdown threshold energy $\beta =E/E_{thr}$.

When an NIR ultrashort laser pulse is tightly focused into a transparent material, two nonlinear optical interactions, multiphoton and cascade ionization, play a main role for creating a spatially confined so-called critical electron density and an optical breakdown [20,21,36,37,38,39]. According to the state-of-the art model, in the ultrashort pulse duration regime < 100 fs, the breakdown process is dominated by multiphoton ionization, whose absorption coefficient is proportional to $I^{k}$, where I is the irradiation intensity and k the multiplicity of the multiphoton process [40]. In this case, the critical free electron density 10^21^ cm^−3^ was achieved before the cascade ionization began to dominate [41,42,43]. As the pulse duration increased, the breakdown process was dominated by cascade ionization. As a consequence, for picosecond pulse durations the critical electron density was achieved much later in the pulse due to the decreasing role of the multiphoton ionization [31,44], thus a higher fraction of the laser pulse energy was transmitted before the breakdown occurred. According to Noack et al., at a wavelength of 1064 nm and for pulse durations between 10 fs and 10 ps, the absorption coefficient in the optical breakdown in water decreases with increasing pulse duration [44].

The pulse duration can only explain the tendency of the absorption value in the optical breakdown. For the total absorption, the ratio of pulse and threshold energy β at a specific pulse duration also has to be taken into account. Below an optical breakdown threshold energy, $E_{thr},$ an NIR laser pulse is mainly transmitted and only slightly attenuated by the relatively low linear absorption coefficient of 0.13 cm^−1^ in water [31]. Above the breakdown threshold, the pulse energy is abruptly absorbed in the breakdown. The absorption increases with the dimensionless parameter $\beta =E/E_{thr}$ [31,45] from typical values of 30% to about 50% for β values ranging from 5 to 20 for a 30 ps laser pulse [31].

In our experiments the measured breakdown threshold $E_{thr}$ increased from 0.12 µJ at 0.6 ps to 0.28 µJ at 14.1 ps (Appendix A) corresponding to β values of 14 and 7, respectively. Our measured $E_{thr}$ were in good agreement with previous work addressing pulse durations increasing from 300 fs to 12 ns [20,45]. Thus, at the constant pulse energy of 2 µJ in our experiments, the dimensionless parameter β decreased with longer pulse durations and led to a 40% decrease of total absorption from about 50% at 0.6 ps to about 30% at 14.1 ps, which may explain, why the bubble energy conversion efficiency also dropped by 40% from 12.5% at 0.6 ps to 7.5% at 14.1 ps.

The pulse-duration-dependent total absorption decrease may be the key reason for the reduction of the bubble energy of about 40% comparing 0.6 ps with 14.1 ps pulse duration. The jet kinetic energy, however, highlights a significantly larger drop of 80% with pulse duration, which may be addressed to fluid dynamic dissipation channels of mechanical energy.

By using 2 µJ laser pulse energy and a focus depth of 52 µm, the jets at all pulse durations developed straight up to the acceptor surface without splashing or spraying. Our earlier work proves that under identical parameters, we may assume that the single-cell-laden jets propagate with an identical jet dynamic to that of the cell-free jets [23]. With the above parameters of 2 µJ pulse energy and 52 µm focus depth single cells can be reliably transferred with very high survival rates after transfer through the entire pulse duration range investigated. The observed cell survival rates 15 min after the transfer agree very well with our previous findings obtained for a laser pulse duration of 0.6 ps [22,23]. It has to be mentioned here, that our determination of a short-term 15 min survival after the transfer cannot replace a future long-term viability study at picosecond pulse durations. However, it is very promising, that in our earlier work the cell survival rate obtained immediately after the transfer showed no significant difference to the cells’ ability to migrate and proliferate on a 66 h time scale [22,23].

## 5. Conclusions

Our time-resolved investigation confirms that with from 600 fs to 14.1 ps increasing laser pulse durations, about 40% less mechanical energy is available for cavitation bubble expansion due to a reduced absorption efficiency in the optical breakdown, leading to less kinetic energy for jet propagation and lower jet velocities. Under appropriate conditions, single cells can be reliably transferred with very high survival rates over the entire pulse duration in the range investigated. The observed cell survival rates agree very well with our previous findings obtained for a laser pulse duration of 0.6 ps.

With our results it seems feasible to extend single-cell bioprinting from femtosecond to picosecond laser pulse durations and to apply CPA-free, more cost efficient, and compact laser sources with pulse durations in the picosecond range for film-free bioprinting and single-cell transfer. Nevertheless, studies of the long-term cell functionality regarding proliferation and differentiation capability are still necessary and will be part of our future work.

## Figures and Tables

**Figure 1 micromachines-12-01172-f001:**
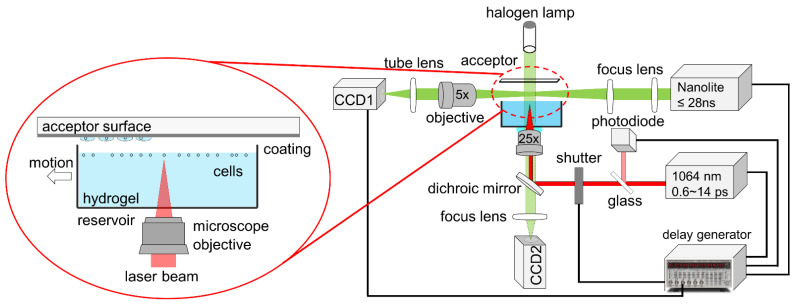
Time-resolved imaging system for visualizing the printing process including cavitation bubble expansion and subsequent jet propagation. The laser beam was focused into the hydrogel by using a water immersion microscope objective. The transfer process was illuminated with a ns white-light pulse and then captured with an imaging system.

**Figure 2 micromachines-12-01172-f002:**
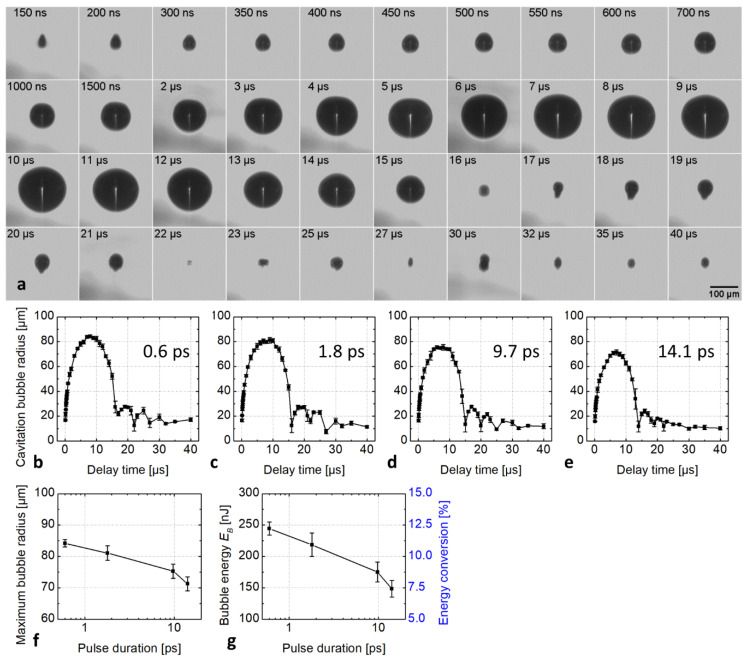
(**a**) Time-resolved images of the cavitation bubble initiated by a 600 fs laser pulse. The laser pulse energy was kept constant at 2.0 µJ and the focus was located 1 mm below the hydrogel surface. The cavitation bubble development at a pulse duration of 1.8, 9.7, and 14.1 ps is similar to that at 600 fs and can be found in Appendix A. (**b**–**e**) Plots of the cavitation bubble radius generated at different pulse duration versus the delay time. (**f**) The plot of the obtained maximum cavitation bubble radius $R_{max}$ in the hydrogel and (**g**) the bubble energy and the corresponding energy conversion efficiency as a function of pulse duration.

**Figure 3 micromachines-12-01172-f003:**
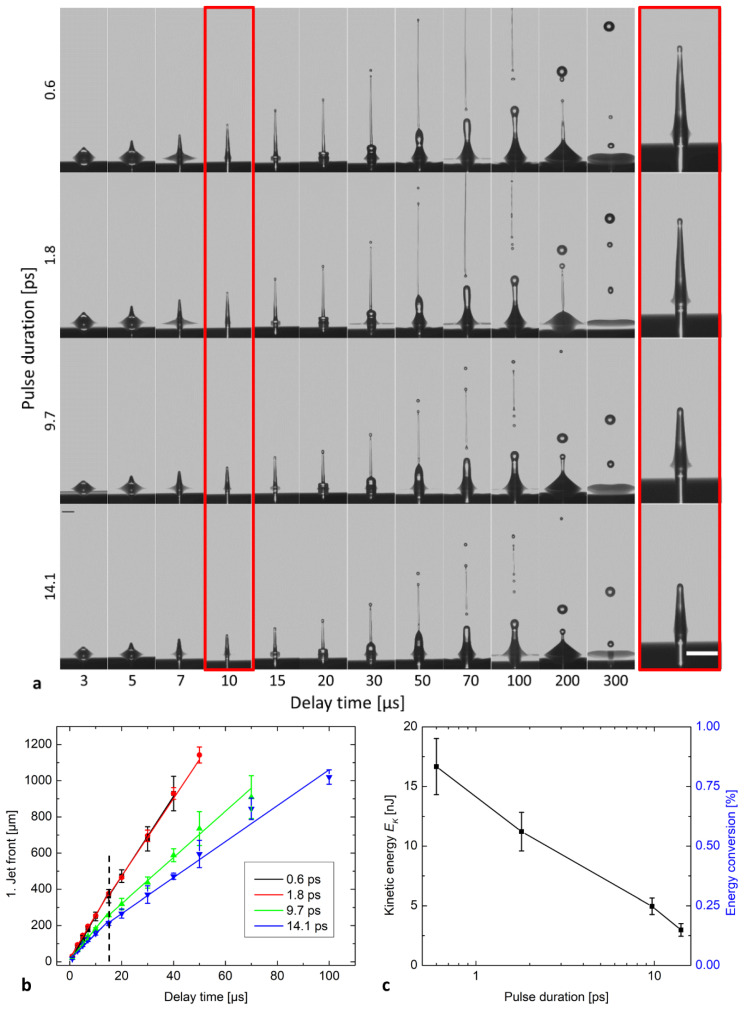
(**a**) Time-resolved images of laser-induced jets in variation of pulse duration, at a constant pulse energy of 2 µJ and focus depth of 52 µm. Scale bar is 100 µm. (**b**) Jet front position versus the delay time at different pulse durations. (**c**) Kinetic energy of hydrogel jets and corresponding conversion efficiency relative to incident pulse energy of 2 µJ at a delay time of 10 µs.

**Figure 4 micromachines-12-01172-f004:**
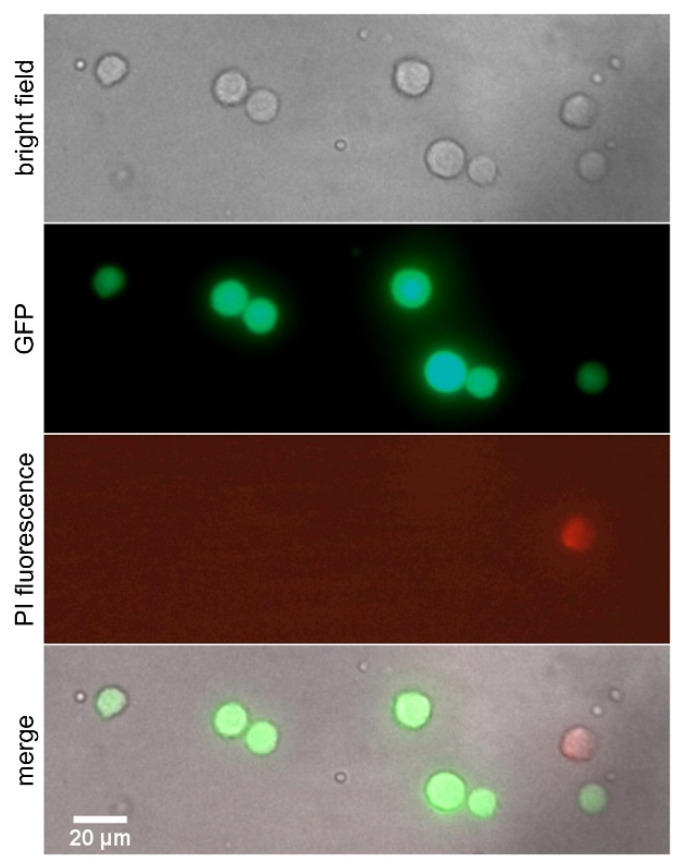
Microscopy images of the transferred hMSC cells on the gelatin-coated acceptor slide after printing. To determine the cell survival rate after transfer, cells were stained with propidium iodide (red PI staining indicates dead cells, live cells are displayed in green).

**Table 1 micromachines-12-01172-t001:** Jet width and velocity of the jets in dependency of the laser pulse duration. Obtained kinetic energy for the jet propagation and corresponding energy conversion efficiency.

τ	d	v	$E_{k}\;(\mathrm{nJ})$	$E_{k}\;(\%)$
0.6	29.6 ± 0.4	25.6 ± 0.6	17 ± 2	0.83 ± 0.1
1.8	28.4 ± 0.3	24.0 ± 0.8	11 ± 2	0.56 ± 0.1
9.7	28.5 ± 0.5	17.5 ± 1.0	5.0 ± 0.7	0.25 ± 0.03
14.1	29.5 ± 1.1	14.3 ± 1.0	3.0 ± 0.5	0.15 ± 0.03

**Table 2 micromachines-12-01172-t002:** Cell survival after transfer of hMSC cells with varying pulse duration.

τ	Transferred Cells	Survived Cells
0.6	18	18
1.8	21	20
9.7	17	17
14.1	17	17

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
