# Peer review of "Extending Single Cell Bioprinting from Femtosecond to Picosecond Laser Pulse Durations"

_micromachines, 2021, doi:10.3390/mi12101172_

Round 1
Reviewer 1 Report
The authors presented using longer pulse duration laser printing for single cells. Compared to traditional fs laser, the proposed ps laser would result in less cost and better integration. However I feel the study still lack of novelty , as laser printing for single cell has been long studied and the results presented in the manuscript are mainly only standard characterization for laser printing, while very little study on the actual factors that affect cell viability are provided, such as stress study, etc. The viability study is also very weak with only live cell numbers, the authors didn't provide any pictures of live cells showing their morphology, nor do they have the transfer efficiency of the single cells. Overall, I feel the manuscript lack of novelty and solid results, and therefore would not recommend this manuscript to be published, unless more in-depth study and lot more results on cell behavior can be provided.
Reviewer 2 Report
The reduction of energy deposited to cells during bioprinting is highly relevant and needs still further optimization to fulfil the demands in future applications in tissue engineering.
The present manuscript “Single-cell bioprinting by using femto- and picosecond laser pulses” describes a method for bioprinting single cells by femto- and picosecond laser pulsed. hMSC were transferred from a reservoir to a acceptance surface by LIFT by varying the laser pulse duration. The authors found that longer laser pulse duration leads to lower energy deposited to the hydrogel and biological systems. The study focus mainly on the technical performance of the system but includes also basic biological experiments to characterise the technology. The following comments should be included in the study to further increase the quality of the manuscript.
- The biological relevance and background should be highlighted more in introduction
- The plane (cover slip? Acceptance surface?) between reservoir and halogen lamp should be labelled in Figure 1 for better understanding. Additionally, a detailed view on the reservoir and acceptance plane with the used media should be added to clarify the conditions for hMSCs
- Line 92f and 97: formatting error
- Line 120: The presented method relies apparently on a preselection of viable cells. The staining of the whole cell population with PI is critical with regard to e. g. clinical applications. The distinction between viable and dead cells can and should be conducted by non-invasive image analysis without DNA intercalating dyes
- Line 124 and 179f: The authors do not provide any information of the time point of viability tests. This information is crucial to assess the cytocompatibility of the system.
- Line 125, Line 179f and Table 2: the population of analysed cells is very small. Due to the heterogeneous characteristics of hMSC populations, more cells (at least 100 cells per replicate should be included in the study).
- Biological replicates are missing: at least three biological replicates (“passages”) must be included for viability test
- The authors should include a fundamental recovery test (viability after e.g. 24h or 48h) to exclude long-term effects triggered by printing. Loss of viability e.g. triggered by mechanical stress leading to apoptosis after several hours can be excluded with such experiments
- Viability after printing and after recovery is only fundamental parameter. In order to apply this method in real workflows, proliferation and differentiaton capability of hMSC must be assessed. The authors should include at least one directed differentiation (e.g. chondrogenic) in their study to prove the functionality of cells after printing.
- The few number of printed cells may lead to the conclusion, that a transfer of the promising technology in real life science applications is challenging/not possible.
- In the current study, an rough estimation of printing time and produced print (dimension,…) must be given.
- The authors should therefore change the focus of their technology to biological cell systems and applications with minor need for upscaling. The generation of meaningful organoids composed of iPS-derived cells could be such an application.
Reviewer 3 Report
Zhang et al. demonstrated the availability of single cell bioprinting by using picosecond laser, which is assumed to be more affordable compared to the femtosecond laser. The conclusions are well-supported by the results and discussion; however, it is believed that the even pico-second laser is still not accessible to majority of researchers, thus the main claim is not convincing enough for readers. To make the logic more persuasive, I would recommend highlighting the scientific advantages of pico-second laser compared to femto-second laser. Additional supporting data should be necessary to support the claim. Below are the additional questions and comments to the authors:
- Is there any specific reason for using the specific hydrogel (i.e. histopaque 1083)?
- No data was found that supports the bio-printability of the developed system (e.g., fluorescence microscope image and others).
Round 2
Reviewer 2 Report
The authors of the manuscript entitled “Extending single cell bioprinting from femtosecond to picosec-2 ond laser pulse durations” addressed all comments of the first review and added also new data.
Especially in the biological part of the manuscript, necessary description of the experimental conditions as well as new data has been added. The authors added missing components to Fig. 1 what clarifies the overall technical setup. The higher number of analysed cells support know the author’s findings. However, the presented technology must be further evaluated regarding the functionality of printed cells (proliferation, differentiation capability). These analyses are fundamental and should be part of future studies.
Author Response
We thank the reviewer for accepting our revised manuscript and his comment on the necessity of future cell functionality studies. Therefore we added this sentence at the end of the conclusion:
"Nevertheless, studies of the long-term cell functionality regarding proliferation and differentiation capability are still necessary and will be part of our future work."
We hope with this change the paper can be accepted for publication.
Reviewer 3 Report
The authors answered all the previous comments, and revised the manuscript well. I believe that the paper is now ready to be published to Micromachines.
Author Response
We thank the reviewer for accepting our revised manuscript for publication.